**A revision of the Combined Drought Indicator (CDI) used in the European Drought**
**Observatory (EDO)**
Carmelo Cammalleri[1,*], Carolina Arias-Muñoz[2], Paulo Barbosa[1], Alfred de Jager[1], Diego Magni[3],
Dario Masante[3], Marco Mazzeschi[4], Niall McCormick[1], Gustavo Naumann[1], Jonathan Spinoni[1] and
Jürgen Vogt[1]
[1] European Commission, Joint Research Centre (JRC), Ispra, Italy.
[2] Arhs Developments, Milan, Italy.
[3] Arcadia SIT, Vigevano, Italy.
[4] UniSystems Luxembourg Sàrl, Luxembourg.
[*] C. Cammalleri (carmelo.cammalleri@ec.europa.eu)

## Abstract

Building on almost ten years of expertise and operational application of the Combined Drought
Indicator (CDI), which is implemented within the European Commission's European Drought
Observatory (EDO) for the purposes of early warning and monitoring of agricultural droughts in
Europe, this paper proposes a revised version of the index. The CDI conceptualizes drought as a
cascade process, where a precipitation shortage (WATCH stage) develops into a soil water deficit
(WARNING stage), which in turn leads to stress for vegetation (ALERT stage). The main goal of the
revised CDI proposed here is to improve the indicator's performance for those events that are
currently not reliably represented, without altering either the modelling conceptual framework or
the required input datasets. This is achieved by means of two main modifications: (a) use of the
previously occurring CDI value to improve the temporal consistency of the time series, (b)
introduction of two temporary classes - namely TEMPORARY RECOVERY for soil moisture and
vegetation greenness, respectively - to avoid brief discontinuities in a stage. The efficacy of the
modifications is tested by comparing the performances of the revised and currently implemented
versions of the indicator for actual drought events in Europe during the last 20 years. The revised
CDI reliably reproduces the evolution of major droughts, out-performing the current version of the
indicator, especially for long-lasting events, and reducing the overall temporal inconsistencies in
stage sequencing of about 70%. Since the revised CDI does not need supplementary input
datasets, it is suitable for operational implementation within the EDO drought monitoring system.
**Keywords:** agricultural drought, SPI, soil moisture, FAPAR, drought monitoring.

## 1. Introduction

In the past 20 years, the monitoring of drought events has gained increasing relevance thanks to
the shift in the paradigm for drought risk management from a reactive to a proactive approach
(Wilhite and Pulwarty, 2005). As advocated by WMO and GWP (2014), drought monitoring and
early warning systems represent one of the three main pillars for successful integrated drought
management (the others being vulnerability and impact assessment, and drought preparedness,
mitigation, and response). A drought monitoring and early warning system identifies climate and
water resources trends and detects the emergence or probability of occurrence and the likely
severity of droughts and its impacts, and should provide reliable information about impending
drought conditions that can be timely communicated to water managers, policy makers, and the
public (Vogt et al., 2018a).

As highlighted in WMO and GWP (2016), monitoring the different aspects of drought may

require a variety of drought indicators and indices. In particular, the authors distinguish among
three typologies of index-based monitoring systems: i) single indicator, ii) multiple indicators, and
iii) composite or hybrid indicators. The latter group allows the integration of a potential large
number of elements into the assessment process of drought characteristics.

A progenitor in the composite indicator category is the approach developed in the United

State Drought Monitor (https://droughtmonitor.unl.edu), based on an expert-supervised
combination of a percentile ranking of several indices for a weekly-based index (Svoboda et al.,
2002). Another combined indicator, which was developed as part of the operational Global
Integrated Drought Monitoring and Prediction System (GIDMaPS, http://drought.eng.uci.edu), is
the Multivariate Standardized Drought Index (MSDI, Hao and AghaKouchak, 2013), which is based
on a combination of soil moisture and precipitation anomalies through a copula function.

At a European scale, the Combined Drought Indicator (CDI) provides a concise

representation of the evolution of agricultural droughts, suitable for communication to both
specialized end-users, policy-makers and the general public (Vogt et al., 2018b). The CDI, originally
conceived by Sepulcre-Canto et al. (2012), has been successfully applied within the European
Drought Observatory (EDO; https://edo.jrc.ec.europa.eu) of the EU's Copernicus Emergency
Management Service (https://emergency.copernicus.eu), as part of a near-real time monitoring
with dekadal (roughly 10 days, 3 times at month) updates and a time-lag of just a few days.

A similar combining approach, albeit with a strong focus on agricultural production and food

security, has been recently implemented as part of the European Commission's Anomaly hot Spots
of Agricultural Production (ASAP, https://mars.jrc.ec.europa.eu/asap) system (Rembold et al.,

2019).

Other hybrid drought indicators, mostly based on the combination of meteorological soil

moisture and streamflow indices via artificial neural networks or entropy theory, were recently
introduced in the literature and applied in several regional studies (i.e. Karamoutz et al., 2009;
Yang et al., 2014; Zhu et al., 2018).

Regarding the CDI, it has proved to be effective at reliably capturing the start and

development of most of the severe droughts that affected European countries throughout almost
10 years of its operational use in EDO, as documented by the analytical drought reports that are
regularly        published        through        the        EDO        web        portal
(https://edo.jrc.ec.europa.eu/edov2/php/index.php?id=1051). Maps of EDO's CDI have also been
extensively used by the European Commission's Emergency Response Coordination Centre (ERCC),
for    their    daily    maps    on    the    most    important    ongoing    emergency    events
(https://erccportal.jrc.ec.europa.eu/Maps/Daily-maps).

While the CDI can claim a considerable number of successful applications in the case of

recognized drought events, a day-by-day analysis of its various components has led to an
increased understanding of its behaviour, and has also highlighted potential improvements,
particularly with regard to its temporal consistency in the case of long-lasting events. The resulting
expertise, which is based on extensive practical experience and a long history of actual cases, can
be used to improve the indicator's performance in those circumstances where it currently may fall
short of expectations. However, given the operational nature of the index, and its reliance on the
availability of near real-time input data, changes on the current forcing data are not considered at
this stage, since this may require the acquisition of additional datasets not readily available in an
operational context. Additionally, any modifications to the modelling framework of an established
indicator such as the CDI, must take into account the existing considerable community of users,
who are accustomed to the indicator in its current form, as well as its acceptance within the
scientific community as a recognized indicator (e.g. Clark et al., 2016; Mariani et al., 2018; WMO
and GWP, 2016), as further exemplified by its use in major case-studies and inter-comparison
analyses (e.g. Blauhut et al., 2016; Jiménez-Donaire et al., 2020; Schwarz et al., 2020).
In light of these considerations, the main goal of this paper is to propose a revised version of
the CDI, with a focus on improving the overall quality of the indicator's performance without
introducing additional or alternative input datasets, and preserving the original modelling concept
that has achieved successful results over many documented case studies. To this end, the study
compares the performance of the proposed revision of the indicator against the current
operational EDO version during some of the main drought events in Europe in the past 20 years.
The spatio-temporal characteristics of these droughts were derived from independent data
sources, such as yield and impacts databases, and were used as reference to assess the
consistency of the model outcomes with the background theoretical framework and the
adherence to the observed real drought dynamics.

**2. Material and Methods**
In this section, the input datasets that are used for computing the CDI are described, and the
computation methods that are applied in both the current version and proposed revision of the
indicator are outlined. The set of case studies of past drought events used to compare the
performances of the current and proposed new versions of the indicator is also described,
together with the adopted evaluation strategy.
**2.1 Input datasets**
The Combined Drought Indicator (CDI) is computed on the basis of the inter-dependency of three
main variables: precipitation, soil moisture, and vegetation greenness. The values for each of these
quantities are standardized as deviations from historical climatology, and compared with a
threshold value to discriminate between normal and extreme conditions. While the data
processing approach is conceptually analogous for all three variables, some peculiarities (for
example regarding the data's spatio-temporal resolution and reference baseline) are worth
highlighting, and these are described in the following sub-sections.
***2.1.1 Precipitation***
Monthly precipitation maps at a spatial resolution of 0.25 degrees are derived by blending daily
rainfall observations at SYNOP (Surface Synoptic Observations) stations from the MARS database
(http://mars.jrc.ec.europa.eu) of the European Commission's Joint Research Centre (JRC), with
monthly precipitation maps at a spatial resolution of 1.0 degree from the Global Precipitation
Climatology Centre (GPCC, http://gpcp.dwd.de).

The 1-month and 3-month Standardized Precipitation Index (SPI-1 and SPI-3, respectively,

McKee et al., 1993) are calculated using the two-parameter gamma distribution fitted over a 30-
year reference period (1981-2010) using the maximum likelihood estimators of Thom (1958) and
Greenwood and Durand (1960). SPI-3 is selected because of its documented correlation with
agricultural drought (WMO, 2012), whereas SPI-1 is selected due to its suitability for detecting the
possible occurrence of flash droughts (when combined with increased evaporative demand due to
high temperatures, low humidity and/or strong winds), as described by Otkin et al. (2018). In line
with Sepulcre-Canto et al. (2012), a threshold value of -1.0 is used for SPI-3, marking the start of
moderately dry conditions according to McKee et al. (1993), whereas a threshold value of -2.0 is
used for SPI-1, denoting the start of extremely dry conditions.

For computing the CDI, both SPI indicators are used jointly to detect precipitation shortages.

Hence, for the sake of simplicity a Boolean SPI indicator (zSPI) is defined, which assumes a value of
1 if either SPI-1 or SPI-3 reports a dry status, as follows:
$$zSPI = \begin{cases} 1 & \text{SPI-3} < -1 \quad or \quad \text{SPI-1} < -2 \\ 0 & otherwise \end{cases}$$      (1)
**_2.1.2 Soil Moisture_**
The soil moisture anomaly index (zSM) is computed using the modelled soil moisture output of the
LISFLOOD hydrological precipitation-runoff model (De Roo et al., 2000). Firstly, dekadal (roughly
10-day) maps of the Soil Moisture Index (SMI; Seneviratne et al., 2010) are computed at a spatial
resolution of 5 km, as a weighted average of the daily volumetric soil moisture values produced by
LISFLOOD for the skin and root zone layers. Successively, the zSM is computed as standardized
deviations (i.e. z-scores) of the values from the full available period (1995-2018).
In the present study, SMI replaces the soil suction (pF) that was previously used both within
EDO and for the original development of the CDI. This has been done as part of a reorganization of
the EDO data portal, in order to improve the readability of maps for non-expert users, given that
SMI simply ranges from 0 (dry) to 1 (wet). Since both SMI and pF are derived from the same daily
volumetric soil moisture dataset and using the same pedotransfer function (PTF; Laguardia and
Niemeyer, 2008), the obtained zSM maps are in practical terms the opposite of the Anomaly pF
used in Sepulcre-Canto et al. (2012). Following these considerations, a threshold of -1 is adopted
to discriminate dry conditions in zSM, analogously to what is used for SPI-3.
**_2.1.3 Vegetation greenness_**
In this study, the biophysical variable Fraction of Absorbed Photosynthetically Active Radiation
(FAPAR), which is estimated from satellite remote sensing data, is used as a proxy for the health
status of vegetation. Sepulcre-Canto et al. (2012) adopted the 10-day composite FAPAR images
provided by the European Space Agency (ESA), derived from the Medium Resolution Imaging
Spectrometer (MERIS) on board of the ENVISAT platform. Following the failure of ENVISAT in 2012,
the MOD15A2H Collection 6 FAPAR product (Myneni, 2015), as derived from the Moderate-
Resolution Imaging Spectroradiometer (MODIS) sensor on board of the Terra satellite, has been
used as a replacement in the operational implementation of the CDI.
The MOD15A2H product is provided by the US National Aeronautics and Space
Administration (NASA) at spatial resolution of 500 metres, as 8-day maximum composites. Within
EDO, these raw data are re-projected onto a 0.01 degrees latitude / longitude regular grid, and
dekadal maps are derived by means of a weighted average of the two closest 8-day maps followed
by an exponential smoothing (Cammalleri et al., 2019). As in the case for soil moisture, anomalies
of FAPAR (zFAPAR) are computed as a standardized z-score on the full available dataset baseline
period (2001-2018). Also here, a threshold value of -1.0 is adopted to highlight dry conditions.
**2.2 The current version of CDI, as implemented in EDO (CDI-v1)**
As is described in detail by Sepulcre-Canto et al. (2012), in the modelling framework of the CDI the
evolution of a drought event is conceptualized by a cause-effect relationship, assuming that a
shortage in precipitation leads to a soil moisture deficit, culminating in reduced vegetation
productivity. In its original form, data for the variables zSPI, zSM and zFAPAR (see above) are used
to characterize three stages of an idealized agricultural drought:
• WATCH, in which the precipitation is below normal (zSPI = 1), and an early warning signal
of a potential drought affecting agriculture can be observed.
• WARNING, when a precipitation deficit propagates in the hydrological cycle and affects soil
water content (zSPI = 1 & zSM < -1).
• ALERT, when the effects of drought become visible as vegetation stress (zSPI = 1 & zFAPAR
< -1).
During the operational implementation of the indicator, two additional recovery stages were
introduced (see https://edo.jrc.ec.europa.eu/factsheets), aimed at better capturing the fade-out
phase of a drought, namely the PARTIAL RECOVERY and FULL RECOVERY stages. In both stages, the
previous month's zSPI ($zSPI_{m-1}$) is introduced to account for the preceding conditions:
•   PARTIAL RECOVERY: zSPI returns to normal values even if vegetation is still negatively
affected ($zSPI_{m-1} = 1$ & $zSPI = 0$ & $zFAPAR < -1$).
•   FULL RECOVERY: Both precipitation and FAPAR return to normal conditions ($zSPI_{m-1} = 1$ &
$zSPI = 0$ & $zFAPAR \geq -1$).
This operational implementation of the index is the one commonly referred to in the
scientific and technical drought literature when CDI is described.
The CDI modelling framework described above is summarised in Fig. 1, where the different
stages of CDI (from WATCH to FULL RECOVERY) are depicted according to the eight cases that can
be obtained by combining the two possible binary states for each of the three main variables (zSPI,
zSM, zFAPAR), as well as a function of $zSPI_{m-1}$.
Due to its operational status, the maps of the CDI that are currently available in EDO are
always processed using data available up to the release date of a new map. For this reason, some
inconsistencies in the reference baseline and actual data (e.g. FAPAR data source) are present in
this operational dataset. For the present study, a self-consistent dataset has been produced by re-
computing the CDI with the best data available at the end of 2018. This dataset (referred to here
as CDI-v1) consists of 648 dekadal maps at 5-km spatial resolution, from January 2001 to
December 2018. In order to compute the CDI at this spatial resolution, the original data for zSPI
and zFAPAR were initially resampled over the zSM grid, using the nearest neighbour and spatial
average procedure, respectively.
**2.3 The revised version of CDI proposed here (CDI-v2)**
In order to better understand the modifications to the CDI that are proposed here, two case
studies where CDI-v1 was not able to capture in full the evolution of the drought, are first
reported.
The original concept behind the CDI assumes the sequential occurrence of extreme
conditions detected by the three constituent indicators (i.e. SPI, soil moisture anomalies, and
FAPAR anomalies). In fact, while Sepulcre-Canto et al. (2012) illustrated the CDI scheme as a
cascade process (see the schematisation in that paper's Fig. 1), its actual implementation can be
seen more in the context of a nested approach, since each successive stage is contained within the
definition of the previous one. This is exemplified by the inclusive nature of the calculation (see
above, where "&" is used in the definition of the classes). This approach can lead to abrupt breaks
in tracking a drought event, when a substantial temporal shift among the three quantities can be
observed.
For example, the plots in Fig. 2 report the time series of SPI-3 (upper panel), zSM (middle
panel) and zFAPAR (lower panel) for a year that includes a drought event in Spain. Dotted vertical
lines demarcate the full span of the drought event. At the top of each plot, a box demarcates the
period when the stage-specific conditions for WATCH, WARNING and ALERT are met. By an *a*
*posteriori* analysis of the event, it is easy to assess a desirable sequence of stages for each dekad,
as reported in the bottom part of the lower plot (i.e. the ideal outcome of a revised CDI, ideally
CDI-v2). However, from the actual sequence of CDI values (CDI-v1) it can be seen that the event is
interrupted in the middle of the soil moisture deficit period due to the return of precipitation to
normal conditions.
A second example is shown in Fig. 3 for a drought event in France, where the time series of
SPI-3, zSM and zFAPAR suggest an extensive period of soil moisture deficit following a
precipitation deficit, which caused a short period of FAPAR anomalies. Even if two periods meeting
the requirement for a WARNING and an ALERT status are observed (see boxes at the top of the
middle and lower panels, respectively), a temporary return above the thresholds is observed (for
one or two dekads) in both zSM and zFAPAR time series. In an *a posteriori* analysis, a single
continuous ALERT period would have been likely detected (see ideal CDI sequence at the bottom
of Fig. 3). CDI-v1 instead treats those gaps as interruptions, causing a back-and-forth transition
between the ALERT and WARNING stages.
This behaviour is in contrast to the cause-effect principle on which the indicator is based,
and even if this occurrence cannot be always avoided in real case studies, it should be kept to a
minimum. It is worth noting how, also in this second case, according to CDI-v1 the event stops well
before the end of the soil moisture deficit, due to the return of precipitation to normal conditions
(SPI-3 > -1).
The two examples reported above highlight the main drawbacks of the current operational
version of the CDI, which can be summarized as follow:
•    Lack of a proper cascade process in favour of a nested approach, which can cause an early
interruption in drought events in case of notable shifts between time series.
•    Absence of a check for possible small gaps within a stage, which can lead to inconsistencies
in the temporal sequence and quick alternation of different stages.
The revised version of the CDI that is proposed here (i.e. hereafter called CDI-v2) addresses
these two key issues by introducing two principal modifications:
•    Set-up of different rules to ensure temporal continuity based on the previous dekad's CDI
($CDI_{d-1}$) rather than the preceding SPI ($SPI_{m-1}$).
•    Addition of a second set of threshold values to detect both temporary gaps within a stage,
and the fade-out phase of a drought.
These modifications are implemented according to the scheme depicted in Fig. 4, where the
upper part of the Table is analogous to that of Fig. 1, while the lower part details the values
assumed by the index for all the possible cases of preceding CDI values.
By juxtaposing Figs. 1 and 4, it is possible to highlight the main changes introduced after
discriminating the outputs on the basis of $CDI_{d-1}$. On the one hand, it is possible to notice how CDI-
v2 (i.e. the proposed revision) behaves identically to CDI-v1 (i.e. the current version) at the start of
a new event (first row, $CDI_{d-1}$ = 0 or 4). On the other hand, for an on-going event ($CDI_{d-1}$ =
1,2,5,3,6), CDI-v2 still behaves similarly to CDI-v1 for the combinations *a-b* and *f-h*, whereas some
major differences can be observed for the cases *c-e*. In these latter instances, both the WARNING
and ALERT stages are preserved if zSM and zFAPAR values support these conditions independently
from the value of zSPI. This modification aims at solving the problem highlighted by the example in
Fig. 2.
The lower part of the table in Fig. 4 highlights how the inclusion of a second threshold for
zSM and zFAPAR (i.e. 0.0 in both cases) aims at addressing those situations when the CDI tends to
return to a stage that conceptually precedes that of the previous dekad (i.e. a WARNING following
an ALERT). In all these circumstances, two TEMPORARY RECOVERY stages are introduced - one for
soil moisture and one for FAPAR - if the values of zSM or zFAPAR fall between the two threshold
values (i.e. -1.0 and 0.0). Since these classes are meant to be temporary, we wish to avoid that the
index remains locked in these classes for long periods. For this reason, a constraint on the
maximum duration of the TEMPORARY RECOVERY stages is fixed at 4 dekads. This value is chosen
as the minimum length to ensure the inclusion of two consecutive monthly zSPI values.
**2.4 Past drought events**
In absence of a reliable independent benchmark for the evaluation of the CDI behaviour, the
performance of the proposed revision of the CDI (CDI-v2 in this paper) is compared against the
current version of the index (called CDI-v1) over selected past drought events in Europe occurring
during the period 2001-2018 (years when all the input datasets are overlapping).
Several drought events of different extent and severity were observed during the reference

period, including the three large-scale and renowned events of 2003 in central Europe (Rebetez et al., 2006),  2005 in Iberia Peninsula (Garcia-Herrera et al., 2007) and 2018 in northern Europe (Buras et al., 2019). Other documented events at national / regional scale include the droughts in Italy and Romania in 2007, western Germany / France in 2011, Romania and Portugal in 2012, eastern Spain in 2014, eastern France / western Germany in 2015 and central Italy in 2017.

For these events, the improvement in the coherence between the proposed revision of the index and the CDI theoretical modelling framework is firstly verified for two test datasets of locations where the operational CDI-v1 was successfully validated in the past. The first dataset of locations corresponds to drought events that were originally used by Sepulcre-Canto et al. (2012) to validate the index. These include data from: Magdeburg (DE), Ciampino (IT) and Wattisham (UK) during the 2003 drought; Albacete (ES) and Beja (PT) in 2005-2004; Ciampino (IT) for the drought in 2007; and Magdeburg (DE) and Deols (FR) during 2011.

The second dataset of locations is derived from the droughts documented in the reports produced by EDO (https://edo.jrc.ec.europa.eu/edov2/php/index.php?id=1051) since the CDI's operational implementation. These include data from: Lisbon (PT) in 2012; Valencia (ES) for the 2014 drought; Strasbourg (FR) in 2015; Rome (IT) during summer 2017; and Dublin (IE), Hannover (DE), Poznan (PL) and Silkeborg (DK) for the drought in 2018.

This qualitative analysis over selected test sites is complemented by a quantitative analysis on the full dataset that evaluates the frequency in which each cell experiences a stage sequencing in contrast with the assumed cause-effect modelling (i.e. a dekad with WARNING followed by one with WATCH), providing a metric to quantify the improvements associated with the proposed revision.

**2.5 Evaluation strategy**

Long records of yield data for cereals (including rice) from the EUROSTAT database were used to

detect specific regions with documented drought impacts in agriculture during the above-reported
drought years. Even if it was not possible to extract evidence of drought impacts for all the events,
mainly due to gaps in data records, six regions were detected from the above-mentioned drought
years, as summarized in Table 1. The reported yield data show how the production was lower than
the long-term average yield for all the regions, as they were actually the minimum in the records
for all the cases, the only exception being ES62, Region of Murcia (which recorded the second to
last yield in 2014 only after 2005).
Assuming that the reduction in yield is a measure of the impacts of drought over vegetated
land, statistics of the ALERT stage in these EUROSTAT NUTS (Nomenclature of Territorial Units for
Statistics) regions during the drought events were investigated as a means of quantifying the
effects of the proposed modification of the CDI. The duration of the drought according to the CDI
is quantified as the period when the percentage of NUTS with WATCH+WARNING+ALERT is at least
20%, and within this period the average percentage of area under ALERT ($P_{ALERT}$) and the maximum
modelled ALERT percentage in the same period ($M_{ALERT}$) are computed for the two CDI versions,
assuming that high values in both $P_{ALERT}$ and $M_{ALERT}$ are expected in these study cases given the
observed drastic reduction in yield.

**3. Results and Discussion**
**3.1 Temporal consistency of drought stages**
Following the modification introduced, one of the main improvements that may be expected in
the revised version of the CDI (CDI-v2) concerns the temporal consistency at the local scale. For
this reason, an initial test was made to compare the temporal behaviour of the current version
(CDI-v1) and proposed revision (CDI-v2) of the indicator, over selected locations in Europe, during
well-documented drought events.
The plots in Figs. 5 and 6 show dekadal time series of CDI-v1 (upper line) and CDI-v2 (lower
line), with the colours corresponding to the classifications in Figs. 1 and 4, respectively. The sites in
Fig. 5 correspond to the locations used for validation by Sepulcre-Canto et al. (2012), whereas the
sites in Fig. 6 were extrapolated from the EDO reports for the most recent drought events.
In all the sites, the start of the drought event coincides for the two versions of the indicator
(CDI-v1 and CDI-v2), as is to be expected given the analogous conditions adopted to define a new
event. Over some sites, the two versions do not differ substantially, as in the case of Wattisham
and Magdeburg (Fig. 5), and Silkeborg and Poznan (Fig. 6), where only minor signs of the issues
highlighted in Figs. 2 and 3 can be observed. In those study sites, the temporal evolution of the
droughts appears to be well reproduced by both versions of the indicator, with the start-, peak-
and end-dates consistent with the scientific literature for the events (Buras et al., 2020; Ciais et al.,
2005; Hanel et al., 2018; Rebetez et al., 2006).
Conversely, the drought development for the sites of Albacete (2005 drought), Ciampino
(2007 drought), Lisbon (2012 drought) and Valencia (2014 drought), differs substantially for the
revised version (CDI-v2) compared with the current version (CDI-v1), with an overall longer
duration and prolonged periods under the WARNING and ALERT stages. The drought events at
those sites are rather similar to what is depicted in Fig. 2, with a long period of soil water deficit
and plant water stress during the whole dry season following a rainfall deficit early in spring and a
hot and dry summers. In these cases, the new version of the index appears to be capable to
capture those instances when a drought is prolonged by higher than normal evaporative demand
even after the rainfall returns to normal. Considering the well documented severity of those
droughts (Garcia-Herrera et al., 2007; MeteoAM, 2007; Spinoni et al., 2015), the behaviour of CDI-
v2 seems to be much more in line with the expected evolution of the droughts.
Finally, for some study cases - specifically Deols (2011 drought), Strasbourg (2015 drought)
and Dublin (2018 drought) - the erratic behaviour of CDI-v1 that is evident later in the event
(similar to the example of Fig. 3), is replaced by a noticeably smoother dynamic in CDI-v2, which is
more in line with both the desirable sequencing of stages and the expected behaviour of a slow-
evolving phenomenon such as drought.
For most of the test sites, the representation of the temporal evolution of the drought
events by CDI-v2 better fits the conceptual "cause-effect" framework of the indicator, by reducing
inconsistent changes in the drought stages. This is quantified by the data reported in Table 2,
where the percentage of cells experiencing one of the three major unexpected stage sequencing is
reported, specifically: i) WATCH following a WARNING, ii) WATCH following an ALERT, or iii)
WARNING following an ALERT. In all three cases the results, expressed as an average percentage
of the area affected by drought (i.e. the sum of all stages excluding FULL RECOVERY), show a
drastic decrease when the CDI-v2 is used instead of CDI-v1. While the reduction occurs for all the
three conditions considered, major improvements can be observed in the reduction of the
instances when a WARNING is followed by a WATCH (4.25% for CDI-v1 compared with 0.88% for
CDI-v2). Overall, the total percentage of inconsistent sequencing is reduced from about 7% for
CDI-v1 to just 2% for CDI-v2, supporting the assumption that the revised indicator (CDI-v2) better
captures the expected evolution of the droughts compared to the current version (CDI-v1) by
minimizing the unexpected behaviours.
The data in Table 3 summarize some key statistics of the ALERT stage over the areas where
significant impact in agricultural production (i.e. yield) were recorded during past droughts (see
Table 1). Overall, both $P_{ALERT}$ and $M_{ALERT}$ are higher for CDI-v2 compared with CDI-v1, with $P_{ALERT}$
being more than double and $M_{ALERT}$ about 30% higher on average for CDI-v2, with the highest
values observed for the two case studies in  Spain and the lowest over  Sweden in 2018. Given the
severe impact of drought over these regions, documented by the concurrent reduced yield
recorded (see Table 1), the large presence of ALERT conditions reported by the CDI-v2 is more in
line with the expected severity of the drought event according to the CDI conceptual modelling
framework.
**3.2 Analysis during major drought events**
An analysis of the full spatio-temporal evolution of the drought events based on the current
(CDI-v1) and revised (CDI-v2) versions of the CDI indicator is performed for the three largest
droughts, as summarised in Figs. 7 to 9 for central Europe (2003), the Iberian Peninsula (2005),
and northern Europe (2018). In each case, the upper plot shows the percentage of the area
affected by drought (i.e. the sum of all stages excluding FULL RECOVERY) for each month, whereas
the maps show examples of the CDI's spatial distribution for selected dekads during the event (as
demarcated by squares on the upper-plot's X-axis).
In all these study cases, it is evident how the percentage of the area that is considered under
drought has a similar temporal behaviour for the two (current and revised) versions of the
indicator, with the latter having only a slightly larger spatial coverage later in the events. An
examination of the maps, however, shows that even if the total area affected is similar, the
partitioning among the different stages may drastically differ around the peak of the drought.
Indeed, the maps for CDI-v1 and CDI-v2 look quite similar at the beginning of the events, but in the
case of CDI-v2 these become much more uniform, and with a higher number of cells under the
ALERT stage, later in the event. The larger number of ALERT in CDI-v2 is more in line with the
conceptualized behaviour of the index, which should reach the ALERT stage at the peak of the
drought development in the case of severe droughts.
The overall dynamic of the 2003 drought (Fig. 7) depicted by the two version of the index is
in line with the historical reconstruction of the event made by the European Drought Impact
Inventory (EDII) and the European Drought Reference (EDR) database
(https://www.geo.uio.no/edc/droughtdb). According to EDII, the event started around April 2003
with a main incidence for eastern Europe up to early June, followed by a propagation through
central Europe and its peak in late August, before ending in November 2003. However, some key
differences in favour of the proposed revision of the index can be observed, such as the higher and
more realistic fraction of areas under ALERT status, which can be seen in the CDI-v2 compared
with CDI-v1 during the drought peak (last map of the series in Fig. 7), against the FULL RECOVERY
areas modelled by CDI-v1 during the expansion of the event in June.

Similarly, the drought event of 2005 over the Iberia Peninsula (Fig. 8) seems to be well

reproduced by both indices. Based on EDII and EDR, the drought in 2005 was part of a longer
drought between autumn / winter 2004 and summer 2006. The event stated in the west,  already
in late 2004, mainly over Portugal, and reached its full extent between July and October 2005, with
a secondary wave observed in summer 2006. The latter was due to the residual deficit that
followed the extremely hot and dry summer of 2005.

This dynamic is well depicted by the plot in Fig. 8 (upper panel), with an already significant

fraction of area under drought at the start of 2005 (about 20% and 30%, according to CDI-v1 and
CDI-v2, respectively) mostly located over  Portugal (see the first map of the series in January
2005). Peak extension is reached in July for the CDI-v1 and between August and September for the
CDI-v2, followed by a slow decline that left still a significant area under drought entering 2006,
especially in the case of CDI-v2. Even if the depiction of the event is quite similar in the first half of
the year (i.e. first three maps of the series), in some circumstances (e.g. between July and August)
the current version (CDI-v1) shows rather different patterns for two consecutive dekads, whereas
the revised version (CDI-v2) gives more temporally consistent outcomes, especially when
comparing maps in succession.

The drought event of 2018 (Fig. 9) was characterized by an extremely warm but not

exceptionally dry spring, that rapidly became an extended and persistent summer drought, due to
the extreme record-breaking temperatures (Peters et al., 2020). This behaviour is well depicted by
both versions of the CDI, with a sudden start between April and June (area under drought jumping
from 0% to 80%) and a quite widespread and enduring drought between July and October. In this
study case, less discrepancies can be observed between the behaviour of the two versions of the
index, compared with the previous two droughts. The most notable difference is the abrupt stop
of drought conditions in Sweden around the peak of the event for CDI-v1 (see last two images of
the series in September and October).
Overall, the analysis of the spatial patterns of both CDI versions during these three major drought
events reveals a more stable behaviour for CDI-v2 compared with CDI-v1. In order to provide a
quantitative estimation of the effects of the proposed changes to the partitioning of drought
stages during an event, the plots of Fig. 10 show the time series of the percentage differences
between CDI-v1 and CDI-v2, in the fraction of the area in the WATCH, WARNING and ALERT stages,
for the same three main droughts that are depicted in Figs. 7-9. Those plots show no substantial
differences at the beginning of each event (first 2-3 months, changes < 5%), and a reduction in the
WATCH fraction for CDI-v2 (negative differences) in favour of an increase in the WARNING and
ALERT fractions during the development of the events. The results are consistent across the three
study cases, suggesting that the revised version of the indicator (CDI-v2) better reflects the "cause-
effect" principle, by showing a progressive propagation of the drought from one stage to the next.
For example, in Fig. 10, some areas that are classified as WATCH by CDI-v1 in a late phase of the
events, are marked as WARNING and ALERT by CDI-v2, with an increased percentage of WARNING
preceding the peak of the drought (June-July in 2003, and May-June in 2018), and an increased
percentage of ALERT at the peak of the event (September in 2003 and 2018; and August-
September in 2005).
It is worth noting that even if some of the largest percentage changes from WATCH to ALERT
occur later in the event (i.e. in autumn after the peak), this is not accompanied by a larger drought
area, as shown by the upper plots of Figs. 7-9. In fact, after the drought has reached its peak, CDI-
v2 depicts an affected area that is reduced in size but mostly constituted by ALERT, whereas in the
previous version WATCH conditions were still reported towards the end of the event.

**4. Summary and Conclusions**
A revised version of the Combined Drought Indicator (CDI), which is currently implemented
operationally within the European Commission's European Drought Observatory (EDO) for
providing early warning and monitoring of agricultural droughts, has been analysed. The proposed
revision of the CDI is based on the extensive experience that has been gained from applying the
indicator during several major drought events that have affected different parts of Europe over
the last ten years.
While the current version of the CDI (called CDI-v1 in this paper) has successfully captured
the onset of most of the documented major drought events, its ability to track correctly the
evolution of events has been limited in the case of long lasting droughts, with significant temporal
shift between reduced rainfall, soil moisture deficit and vegetation stress periods caused by high
temperature and evaporative demand following the rainfall deficit. The proposed revision of the
CDI (called CDI-v2 in this paper) aims at addressing those shortcomings, without either modifying
the required input data or substantially altering the conceptual "cause-effect" framework
underlying its original development, especially given the indicator's proven reliability based on
many case studies and inter-comparison analyses. This enables the retroactive application of the
revised indicator to past drought events, without the need for additional inputs or changes in the
underlying datasets. For similar reasons, the three main stages of drought (i.e. WATCH, WARNING
and ALERT), which were originally defined in Sepulcre-Canto et al. (2012), remain unchanged, as
does the inclusion of a FULL RECOVERY stage to identify the end of a drought period and the
return to normal conditions.

The two main changes that are introduced in the CDI-v2 are:

•   The inclusion of a constraint on the temporal consistency, based on the CDI's value in the
preceding dekad (thus rendering obsolete the previously defined PARTIAL RECOVERY stage).
•   The addition of two TEMPORARY RECOVERY stages - one for soil moisture and the other for
vegetation greenness (represented by FAPAR) - with the aim of improving temporal continuity in
the case of small gaps in the middle of periods that are otherwise characterised by the same
drought stage.

A comparison of the performance of the current version (CDI-v1) and proposed revision

(CDI-v2) of the indicator highlights the capability of CDI-v2 to improve on the results of CDI-v1 in
several circumstances, without impairing the overall performance for drought events that are
already correctly reproduced by CDI-v1. This is indicated by the reduced number of instances
where a specific stage is followed by another that is not coherent with the cause-effect modelling
framework, as well as by the increase in the extension of ALERT areas (i.e. visible vegetation
stress) during events with recorded impacts in agricultural production quantified by reduced
annual yield.

While for a few test cases (e.g. the 2018 drought in northern Europe) only marginal changes

are observed, in the majority of the cases the new version of the indicator (CDI-v2) clearly
outperforms the current version, with an overall better temporal consistency and a more
continuous sequencing of the drought stages. In all the observed study cases, the CDI-v2 returns a
reduced number of cells under WATCH around the peak of the drought in favour of WARNING
(before the peak) and ALERT (at the peak) stages.
On a general level, it is clear that the new version of the indicator better approximates the
expected spatio-temporal characteristics of a drought event in all the performed analyses, with a
more realistic succession of the WATCH, WARNING and ALERT stages, and a large spatial
consistency in the modelled patterns. In addition, in spite of the improved performance of the
revised version of the CDI, the indicator's "look and feel" are not substantially altered. Given the
well established community of users of the current version of the CDI that is implemented in EDO,
this is a key consideration that can ensure a smooth future transition to the operational use within
EDO of the revised version of the CDI that is proposed here.
Finally, with regard to potential further developments of the methodology, in the framework
of the continuous maintenance of the EDO system additional analyses shall be carried out in order
to evaluate the potential integration of other indicators, aimed at better capturing drought events
at different time scales (e.g. indices based on ground water), or to incorporate also information on
evaporative demand into the modelling of meteorological conditions.

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

**Table 1.** Cereals (including rice) yield (t/ha) data for different NUTS regions as derived from the
EUROSTAT database. The column "avg. 2000-2018" reports the average yield during the
whole period, whereas the column "drought year" reports the actual yield for the drought
year specified in the "year" column.

| NUTS | Name | year | yield (t/ha) | |
|------|------|------|--------------|---|
| | | | avg. 2000-2018 | drought year |
| DE1 | Baden-Württemberg | 2003 | 6.8 | 5.7 |
| ES42 | Castile – La Mancha | 2005 | 2.7 | 1.3 |
| RO31 | Sud – Muntenia | 2007 | 3.5 | 2.3 |
| RO12 | Centru | 2012 | 3.4 | 1.1 |
| ES62 | Region of Murcia | 2014 | 1.1 | 0.5 |
| SE21 | Småland | 2018 | 4.3 | 2.9 |


**Table 2.** Average percentage of cells in drought areas with sequencing in contrast with the "cause-
effect" relationship for the full European domain.

| Version | WARNING to WATCH | ALERT to WATCH | ALERT to WARNING |
|---------|------------------|----------------|------------------|
| CDI-v1 | 4.25 | 1.79 | 1.20 |
| CDI-v2 | 0.88 | 0.52 | 0.82 |










**Table 3.** ALERT stage statistics over the NUTS regions with observed yield impacts during drought
events (see table 1). $P_{ALERT}$ is the average percentage of ALERT during the drought duration,
and $M_{ALERT}$ is the maximum percentage in the same period. The drought duration is defined
as the period when the percentage of the NUTS with WATCH+WARNING+ALERT is > 20% for
either CDI-v1 or CDI-v2.

| NUTS | Period | duration (month) | CDI-v1 | | CDI-v2 | |
|---|---|---|---|---|---|---|
| | | | $P_{ALERT}$ | $M_{ALERT}$ | $P_{ALERT}$ | $M_{ALERT}$ |
| DE1 | 1/2003 – 12/2003 | 9 | 12.4 | 70.4 | 25.9 | 79.5 |
| ES42 | 7/2004 – 6/2006 | 16 | 18.9 | 73.6 | 42.8 | 88.5 |
| RO31 | 1/2007 – 12/2007 | 5 | 20.3 | 44.9 | 41.2 | 71.4 |
| RO12 | 9/2011 – 12/2012 | 13 | 5.9 | 36.9 | 17.3 | 45.5 |
| ES62 | 1/2014 – 12/2014 | 10 | 10.2 | 78.2 | 31.8 | 83.0 |
| SE21 | 1/2018 – 12/2018 | 5 | 4.3 | 10.8 | 8.1 | 18.8 |

| | a | b | c | d | e | f | g | h |
|---|---|---|---|---|---|---|---|---|
| zSPI | = 0 | = 1 | = 0 | = 0 | = 0 | = 1 | = 1 | = 1 |
| zSM | ≥ -1 | ≥ -1 | < -1 | ≥ -1 | < -1 | < -1 | ≥ -1 | < -1 |
| zfAPAR | ≥ -1 | ≥ -1 | ≥ -1 | < -1 | < -1 | ≥ -1 | < -1 | < -1 |
| $zSPI_{m-1} = 0$ | 0 | 1 | 0 | 0 | 0 | 2 | 3 | 3 |
| $zSPI_{m-1} = 1$ | 4 | 1 | 4 | 5 | 5 | 2 | 3 | 3 |

| 1 | WATCH | 2 | WARNING | 3 | ALERT | 4 | FULL RECOVERY | 5 | PARTIAL RECOVERY |
|---|---|---|---|---|---|---|---|---|---|


**Figure 1.** Schematic representation of the CDI-v1 computation procedure. The upper part of the table reports the eight possible combinations of the three main Boolean quantities (from *a* to *h*). The lower part of the table reports the corresponding CDI classes for the two possible cases of antecedent zSPI (subscript m-1).

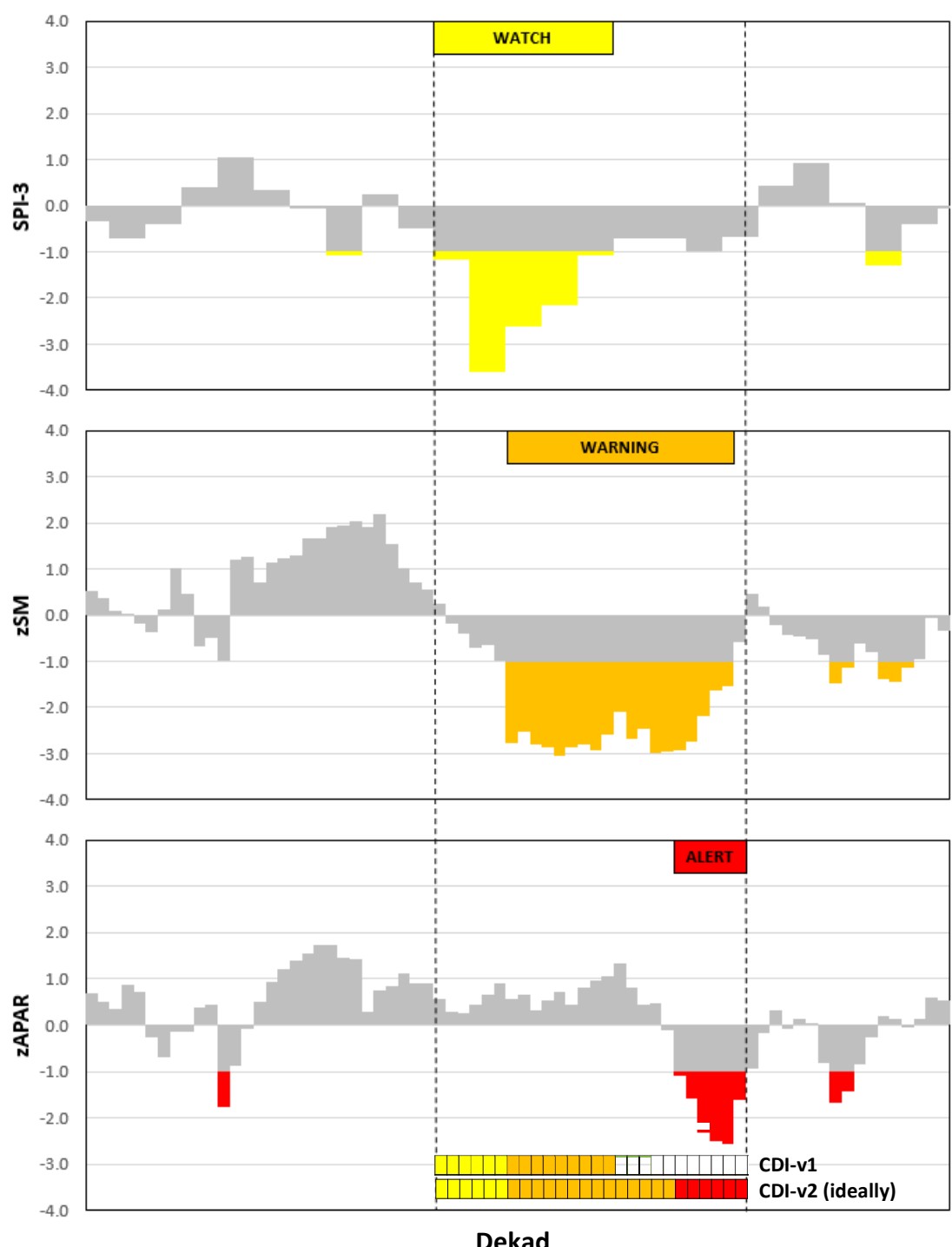

650

**Figure 2.** Example of the possible cascade process driving the evolution in a case of a drought event in Spain. Dotted lines delimit the period under drought, whereas the squares at the bottom of the plots report the outcome of the operational CDI (CDI-v1, upper line) and the ideal evolution of a revised version (CDI-v2 ideally, lower line) values for each dekad.

655

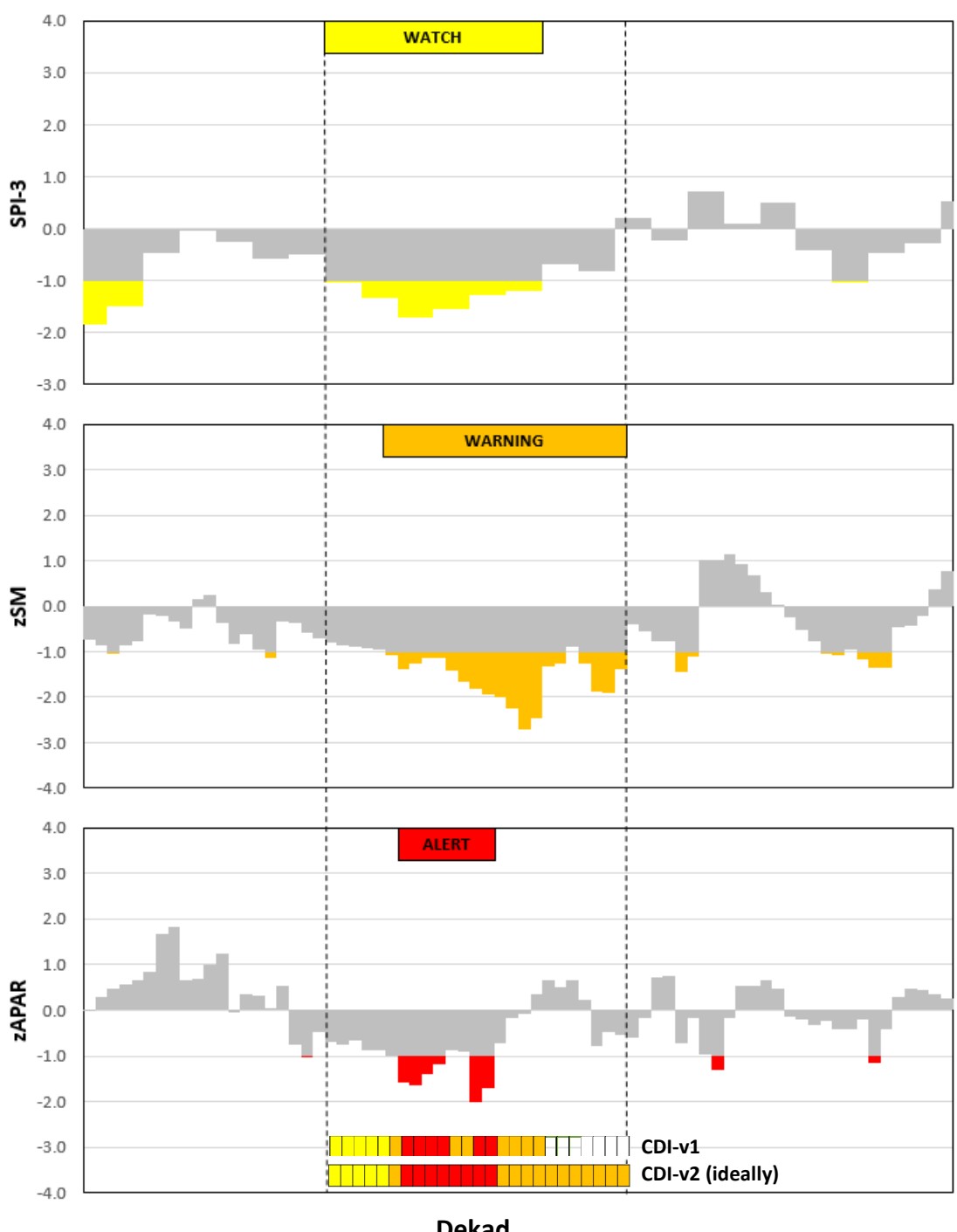

656

**Figure 3.** Example of the small gaps that can occur during a drought event in France. Dotted lines delimit the period under drought, whereas the squares at the bottom of the plots report the outcome of the operational CDI (CDI-v1, upper line) and the ideal evolution of a revised version (CDI-v2 ideally, lower line) values for each dekad.

| | a | b | | c | | d | e | f | | g | h |
|---|---|---|---|---|---|---|---|---|---|---|---|
| zSPI | = 0 | = 1 | | = 0 | | = 0 | = 0 | = 1 | | = 1 | = 1 |
| zSM | ≥ -1 | ≥ -1 | | < -1 | | ≥ -1 | < -1 | < -1 | | ≥ -1 | < -1 |
| | < 0 / ≥ 0 | < 0 / ≥ 0 | | | | | | | | | |
| zfAPAR | ≥ -1 | ≥ -1 | | ≥ -1 | | < -1 | < -1 | ≥ -1 | | < -1 | < -1 |
| | < 0 / ≥ 0 | < 0 / ≥ 0 | | < 0 / ≥ 0 | | | | < 0 / ≥ 0 | | | |
| $CDI_{d-1}$ = 0,4 | 0 | 1 | | 0 | | | | 2 | | 3 | |
| $CDI_{d-1}$ = 1 | 4 | | | | | | | | | | |
| $CDI_{d-1}$ = 2,5 | 5 / 4 | 5 / 1 | | 2 | | 3 | | | | | |
| $CDI_{d-1}$ = 3,6 | 6 / 4 | 6 / 1 | | 6 / 2 | | | | 6 / 2 | | | |

| 1 WATCH | 2 WARNING | 3 ALERT | 4 FULL RECOVERY | 5 TEMP. SM RECOVERY | 6 TEMP. fAPAR RECOVERY |
|---|---|---|---|---|---|

**Figure 4.** Schematic representation of the CDI-v2 computation procedure. The upper part of the table reports the eight possible combinations of the three main Boolean quantities (from *a* to *h*), with sub-cases (based on the second set of thresholds) reported where used. The lower part of the table reports the corresponding CDI classes for all the antecedent CDI values (subscript d-1).

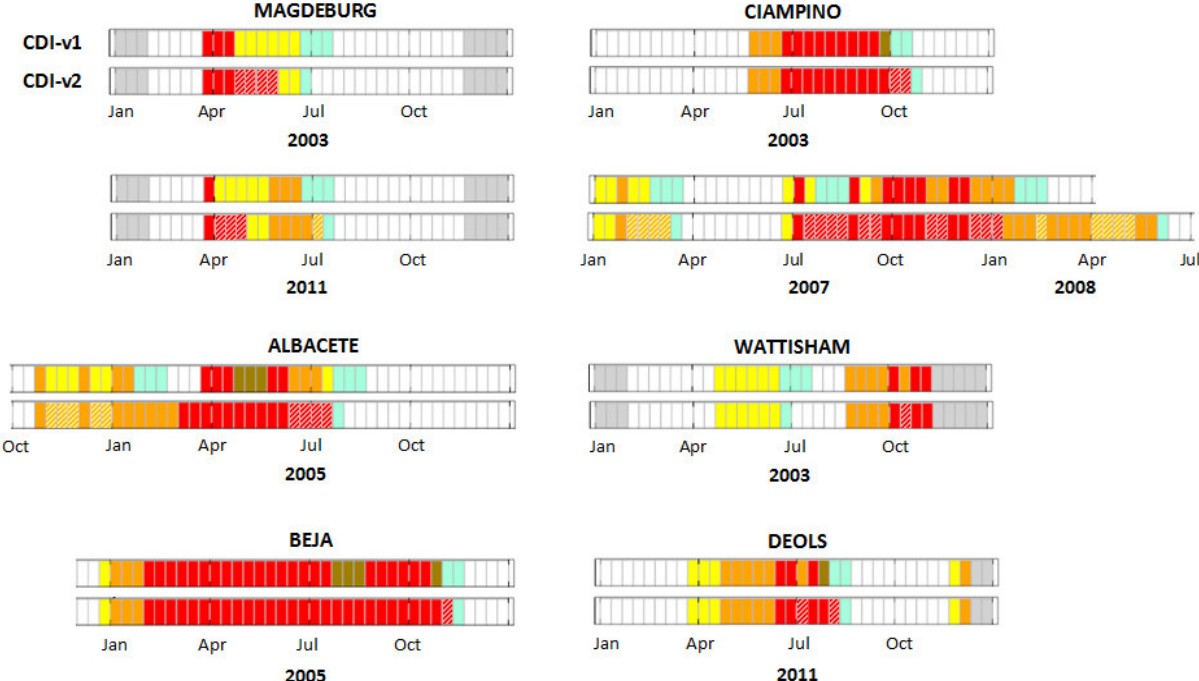

667

**Figure 5.** Time series of CDI-v1 (upper lines) and CDI-v2 (lower lines) for different test sites under

drought between 2001 and 2011, as documented in Sepulcre-Canto et al. (2012). See Figs. 1 and 4

for the corresponding legends. The labels in the x-axis correspond to the beginning of the month.

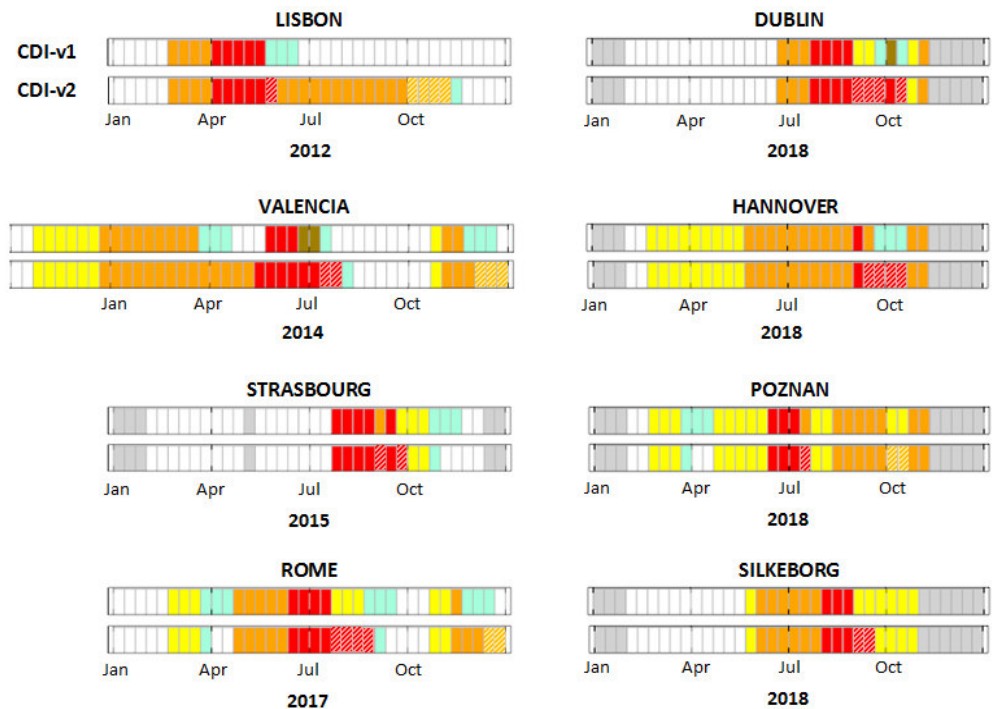

671

**Figure 6.** Time series of CDI-v1 (upper lines) and CDI-v2 (lower lines) for different test sites under
drought between 2012 and 2018, as documented in the analytical drought reports in EDO
(https://edo.jrc.ec.europa.eu/edov2/php/index.php?id=1051). See Figs. 1 and 4 for the
corresponding legends. The labels in the x-axis correspond to the beginning of the month.



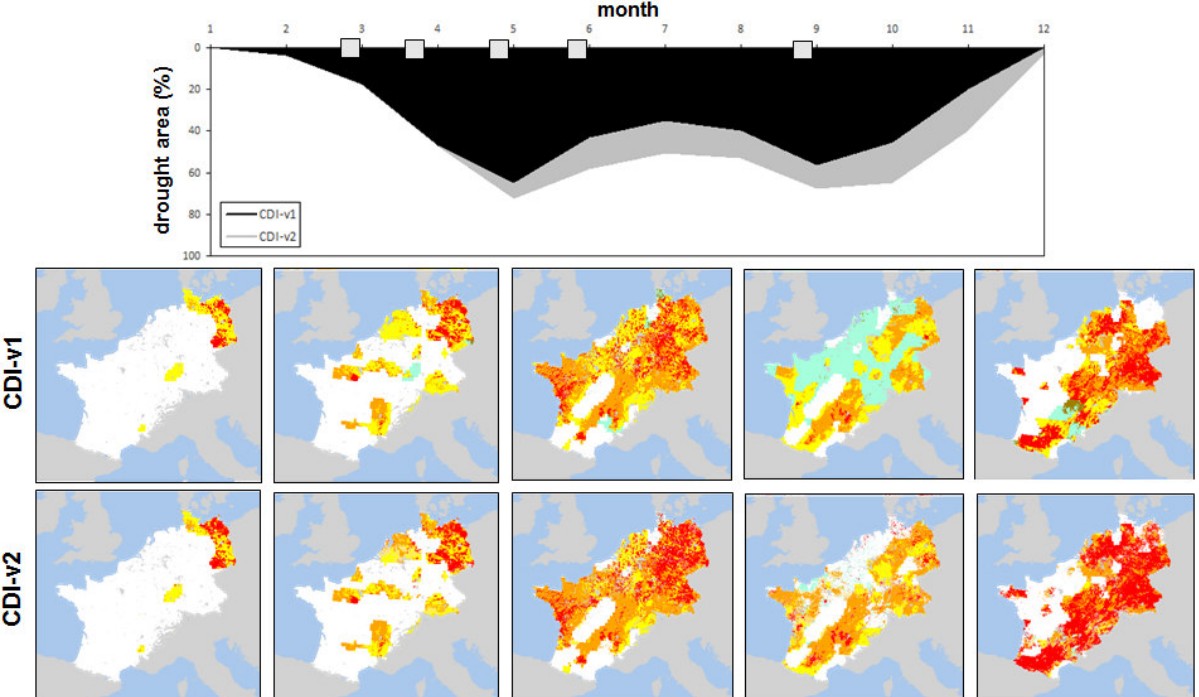

**Figure 7.** Temporal evolution of the 2003 central Europe drought according to the two versions of
the CDI. The upper plot shows the percentage of the area under drought
(WATCH+WARNING+ALERT, in black for CDI-v1 and in grey for CDI-v2), whereas the lower images
depict the spatial distribution of the CDI-v1 (upper row) and CDI-v2 (lower row) for the selected
dekads (demarked in the upper plot by the squares on the x-axis).

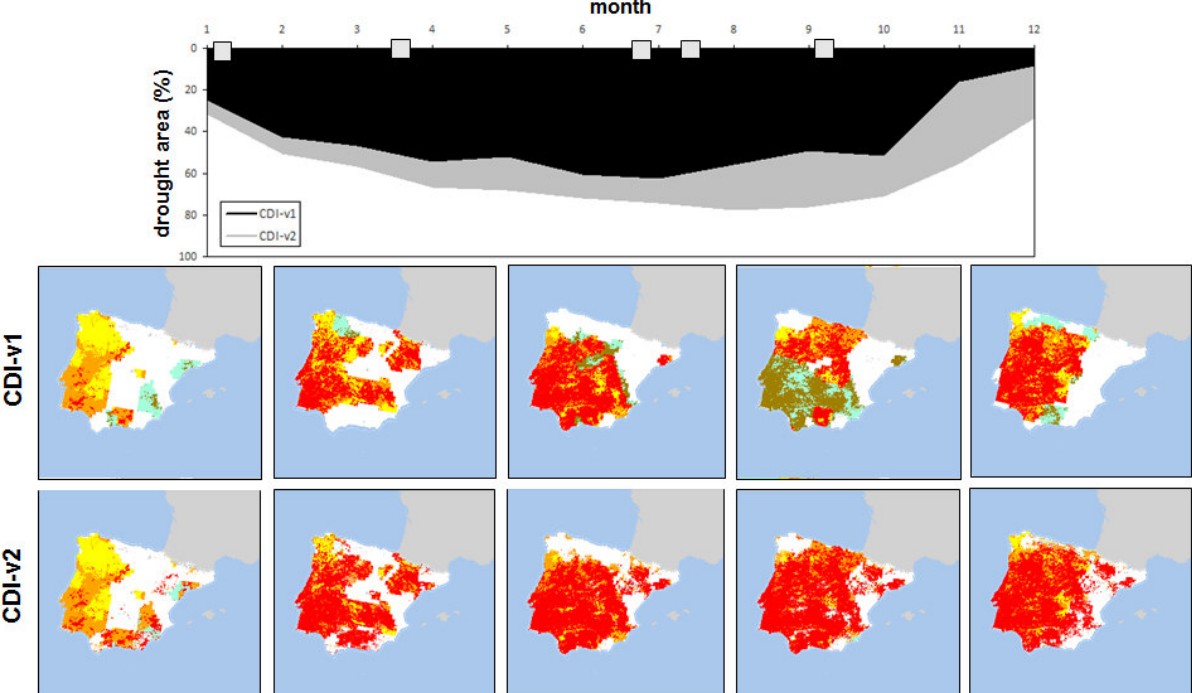


**Figure 8.** Temporal evolution of the 2005 Iberian Peninsula drought according to the two versions

of the CDI. The upper plot shows the percentage of the area under drought

(WATCH+WARNING+ALERT, in black for CDI-v1 and in grey for CDI-v2), whereas the lower images

depict the spatial distribution of the CDI-v1 (upper row) and CDI-v2 (lower row) for the selected

dekads (demarked in the upper plot by the squares on the x-axis).

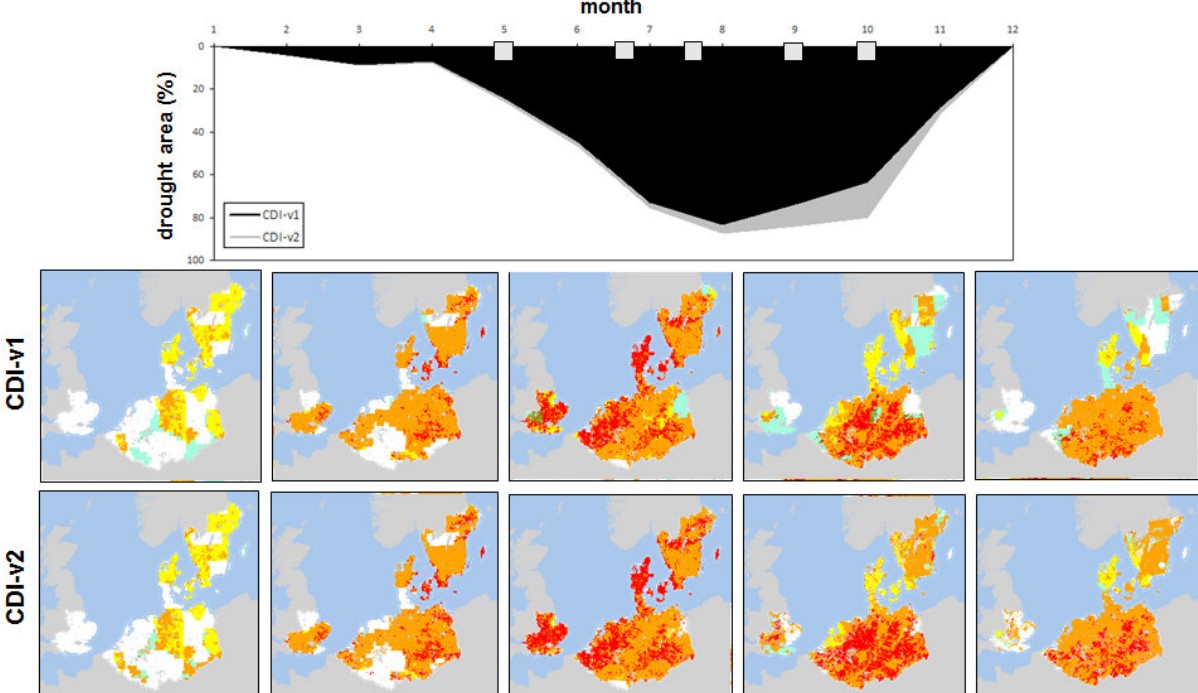


**Figure 9.** Temporal evolution of the 2018 northern Europe drought according to the two versions
of the CDI. The upper plot shows the percentage of the area under drought
(WATCH+WARNING+ALERT, in black for CDI-v1 and in grey for CDI-v2), whereas the lower images
depict the spatial distribution of the CDI-v1 (upper row) and CDI-v2 (lower row) for the selected
dekads (demarked in the upper plot by the squares on the x-axis).

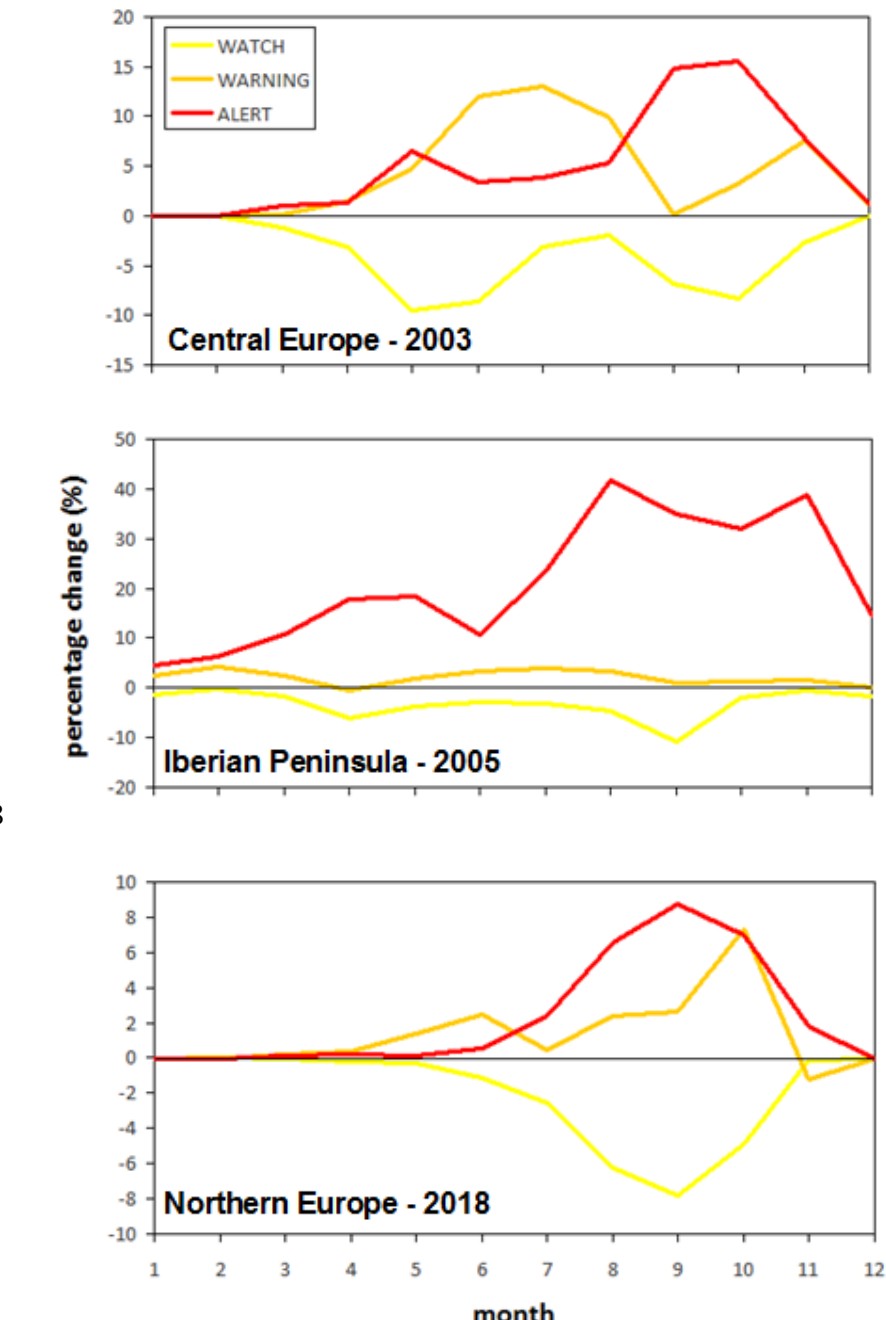



**Figure 10.** Percentage differences between CDI-v1 and CDI-v2 fraction of area in WATCH (yellow line), WARNING (orange line) and ALERT (red line) stages for the same three main droughts depicted in Figs. 7-9. Negative (positive) values indicate a reduction (increase) in the CDI-v2 compared to CDI-v1.