# Peer review of "A revision of the Combined Drought Indicator (CDI) as part of the European"

_Natural Hazards and Earth System Sciences, 2020_

## Referee Comment (RC1) · Anonymous Referee #1 · 15 Sep 2020

The article presents a modification in the way to compute the Combined Drought Index that serves to feed the European Drought Observatory. I will not enter into discussion about the CDI itself. I think it has some flaws in terms of flexibility to deal with environments of different characteristics that respond at different time scales to droughts, and I also prefer the inclusion of the evaporative demand into the computation of the climatic drought index. Said that I also see the value of this index and that is widely accepted, and used as reference for EDO. The proposed modification is rather logical and it implies an improvement in the capability of the index to deal with reversal conditions happening during the evolution of specific drought events. This is why I think that is good to publish the paper, in order to inform to potential users on the characteristics

of the modified index. About the paper itself, I do not have much specific comments to provide, the objective is clear, and it is well structured and written. I would suggest a more critical introduction of the CDI compared to other drought index implemented in monitored systems at large scales, and I would also try to perform a more quantitative assessment of the improvement associated to the modified CDI index, in the current manuscript is merely descriptive (it is true that the case studies suggest a certain improvement compared to CDI-1). The Figures showing the area affected by drought under CDI1 and CDI2 should present labels in their axis to facilitate the reading.

---

## Referee Comment (RC2) · Anonymous Referee #2 · 28 Sep 2020

Review of manuscript "A REVISION OF THE COMBINED DROUGHT INDICATOR (CDI) AS PART OF THE EUROPEAN DROUGHT OBSERVATORY (EDO) by Carmelo Cammalleri, Carolina Arias-Muñoz, Paulo Barbosa, Alfred de Jager, Diego Magni, Dario Masante, Marco Mazzeschi, Niall McCormick, Gustavo Naumann, Jonathan Spinoni and Jürgen Vogt

This manuscript aims to propose and evaluate the new version of the existent Combined Drought Indicator (CDI), implemented at operational way within the European Commission's European Drought Observatory (EDO). The revised CDI aims to better represent a set of events that are currently not reliably represented. In this manuscript,

the authors proposed two main changes to the current CDI and they aim to show the ability of the revised CDI to reproduce major drought evolutionÂňÂň, in particular for long lasting events. The CDI performance was tested by comparison with the current version of the index, considering 4 significant events of the last 2 decades. The overall context of the subject seems to be appropriate for this journal. Despite the crucial role of this type of indices for operational processes, the paper has a very marked technical character, as only shows impacts of the two modifications on the new version of CDI and lacks comparison with other (hybrid or not) indices. Therefore, I consider that this paper could be published in Natural Hazards and earth System Sciences after the authors considering my next comments. 1. Introduction The introduction is short and based in a short number of papers, some of them from co-authors, being based mainly on information of the current CDI. As said before the technical character of the manuscript and the absence of the most recent state of art on drought studies is a caveat of this manuscript. Several recent indices were proposed aiming to include the evaporative demand of vegetation. The importance of these type of drought indicators and their possible inclusion on CDI may be included. 2. Writing and Figure of the manuscript The paper is very descriptive, and the reading is sometimes monotonous. The manuscript is based on several schematic figures, with not very distinguishable colours, namely for black and white versions. Numbers in Figure 5, 6 and 11 are very small. 3. Danger Levels Figures 7 to 10 highlight the increasing of area affected by drought in ALERT stage. Is this realistic? In particular in case of 2003, 2005 and 2018 the increase of ALERT stage area is obvious in fall (Figure 11). Why? The increase of area affected by ALERT stage seems to be compensated by the decrease of area affect by WATCH stage in the case of 2003, 2011 and 2008. However in 2005 a strong increase of ALERT stage is observed in fall, but this is not compensated by the decrease of the other stages. Why? Is this a realistic feature? As far as I know the drought event of 2005 in Iberia started in November 2004 and is ending in summer 2005. 4. Comparison with other hybrid indices In the case of drought is difficult to know when an event starts or ends. The classification of drought is also a challenging task.

Therefore, a validation of CDI or another drought indicator is challenging. However, in my opinion it is not enough to evaluate an indicator without an exhaustive comparison with other indicators (multiscalar indicators, vegetation indicators, among others). A comparison of the new version with the previous version of the same index seems to be not sufficient, namely in the case of a product that is produced and disseminate operationally.
* * *

---

## Author Comment (AC1) · 5 Oct 2020

**Reviewer #1**

The article presents a modification in the way to compute the Combined Drought Index that serves to feed the European Drought Observatory. I will not enter into discussion about the CDI itself. I think it has some flaws in terms of flexibility to deal with environments of different characteristics that respond at different time scales to droughts, and I also prefer the inclusion of the evaporative demand into the computation of the climatic drought index. Said that I also see the value of this index and that is widely accepted, and used as reference for EDO.

We thank the reviewer for this thoughtful comment. As stated in the comment, here we focused solely on a revision of the structure of the CDI, without altering the forcing input datasets. We agree that exploring other base indicators (e.g. SPEI rather than SPI) and temporal scales may lead to further improved performances, but we believe that such modifications should be addressed separately in order to quantify the added value of each change. We plan to investigate further some possible improvements of the index, and we will add considerations on potential further analyses in the conclusions of the revised manuscript.

The proposed modification is rather logical and it implies an improvement in the capability of the index to deal with reversal conditions happening during the evolution of specific drought events. This is why I think that is good to publish the paper, in order to inform to potential users on the characteristics of the modified index.

We appreciate the support shown to the proposed revision of the index.

About the paper itself, I do not have much specific comments to provide, the objective is clear, and it is well structured and written. I would suggest a more critical introduction of the CDI compared to other drought index implemented in monitored systems at large scales,

We will expand the introduction to include references to similar indices including those based on hybrid and combined approaches.

and I would also try to perform a more quantitative assessment of the improvement associated to the modified CDI index, in the current manuscript is merely descriptive (it is true that the case studies suggest a certain improvement compared to CDI-1).

We thank the reviewer for this suggestion. As can be imagined, it is not straightforward to quantify the improvement for the revised index, due to the lack of an obvious reference dataset for the evolution of a past drought event. This is why we focused more on qualitative considerations, and we quantified the improvements (see Table 1) only in terms of number of inconsistencies from the theoretical framework.

Nonetheless we will try to improve the quantitative analysis, by exploring the relationship with other independent sources of data related to agricultural drought impacts.

The Figures showing the area affected by drought under CDI1 and CDI2 should present labels in their axis to facilitate the reading.

We will revise the Figure(s) to improve the readability.

---

## Author Response (AR1)

**Reviewer #1**

The article presents a modification in the way to compute the Combined Drought Index that serves to feed the European Drought Observatory. I will not enter into discussion about the CDI itself. I think it has some flaws in terms of flexibility to deal with environments of different characteristics that respond at different time scales to droughts, and I also prefer the inclusion of the evaporative demand into the computation of the climatic drought index. Said that I also see the value of this index and that is widely accepted, and used as reference for EDO.

We thank the reviewer for his/her thoughtful comments. We revised the introduction to further highlight how the focus of the paper is the revision of the index structure without altering the forcing input datasets. We also added some final considerations on potential further analyses in the conclusions.

The proposed modification is rather logical and it implies an improvement in the capability of the index to deal with reversal conditions happening during the evolution of specific drought events. This is why I think that is good to publish the paper, in order to inform to potential users on the characteristics of the modified index.

We appreciate the support shown to the proposed revision of the index.

About the paper itself, I do not have much specific comments to provide, the objective is clear, and it is well structured and written. I would suggest a more critical introduction of the CDI compared to other drought index implemented in monitored systems at large scales,

We expanded the introduction to include references to other hybrid and combined indices.

and I would also try to perform a more quantitative assessment of the improvement associated to the modified CDI index, in the current manuscript is merely descriptive (it is true that the case studies suggest a certain improvement compared to CDI-1).

We revised the section to better highlighted how the evaluation strategy has the goal of highlighting the improvements of the new version compared to the previous one, rather than a strict validation of the index.

We also added a new section on the characterization of the test drought events, by adding some more quantitative information based on EUROSTAT yield data. These independent information were used to further support the increasing frequency of ALERT stages during the drought events with documented impacts on agricultural production.

The Figures showing the area affected by drought under CDI1 and CDI2 should present labels in their axis to facilitate the reading.

We revised the Figures to improve the readability.

**Reviewer #2**

Review of manuscript "A REVISION OF THE COMBINED DROUGHT INDICATOR (CDI) AS PART OF THE EUROPEAN DROUGHT OBSERVATORY (EDO) by Carmelo Cammalleri, Carolina Arias-Muñoz, Paulo Barbosa, Alfred de Jager, Diego

Magni, Dario Masante, Marco Mazzeschi, Niall McCormick, Gustavo Naumann, Jonathan Spinoni and Jürgen Vogt

This manuscript aims to propose and evaluate the new version of the existent Combined Drought Indicator (CDI), implemented at operational way within the European Commission's European Drought Observatory (EDO). The revised CDI aims to better represent a set of events that are currently not reliably represented. In this manuscript, the authors proposed two main changes to the current CDI and they aim to show the ability of the revised CDI to reproduce major drought evolution, in particular for long lasting events. The CDI performance was tested by comparison with the current version of the index, considering 4 significant events of the last 2 decades. The overall context of the subject seems to be appropriate for this journal. Despite the crucial role of this type of indices for operational processes, the paper has a very marked technical character, as only shows impacts of the two modifications on the new version of CDI and lacks comparison with other (hybrid or not) indices. Therefore, I consider that this paper could be published in Natural Hazards and earth System Sciences after the authors considering my next comments.

We thank the reviewer for his/her comments. We revised the text to address the main comments.

**1. Introduction**

The introduction is short and based in a short number of papers, some of them from co-authors, being based mainly on information of the current CDI. As said before the technical character of the manuscript and the absence of the most recent state of art on drought studies is a caveat of this manuscript. Several recent indices were proposed aiming to include the evaporative demand of vegetation. The importance of these type of drought indicators and their possible inclusion on CDI may be included.

We expanded the introduction to include reference to other hybrid and combined indicators. We also added additional clarifications on how the paper focuses only on revisiting the structure of the index, without altering the input datasets.

**2. Writing and Figure of the manuscript**

The paper is very descriptive, and the reading is sometimes monotonous. The manuscript is based on several schematic figures, with not very distinguishable colours, namely for black and white versions. Numbers in Figure 5, 6 and 11 are very small.

We revised the results and discussion section, by splitting into two sub-sections. We hope that this new format give more structure to the text and guidance to the readers.

The colour schemes used in the Figures are in line with those currently used in the operational EDO system. We think that keeping these schemes consistent is important for readers. The readability of the above mentioned figures was improved by increasing the font size and re-arranging the panels.

**3. Danger Levels**

Figures 7 to 10 highlight the increasing of area affected by drought in ALERT stage. Is this realistic? In particular in case of 2003, 2005 and 2018 the increase of ALERT stage area is obvious in fall (Figure 11). Why? The increase of area affected by ALERT stage seems to be compensated by the decrease of area affect by WATCH stage in the case of 2003, 2011 and 2008. However in 2005 a strong increase of ALERT stage is observed in fall, but this is not compensated by the decrease of the other stages. Why? Is this a realistic feature? As far as I know the drought event of 2005 in Iberia started in November 2004 and is ending in summer 2005.

The increase in area for ALERT observed in the later stage of the droughts (peak and after) is realistic if we follow the assumption that drought propagates from rainfall to soil moisture to vegetation, as conceptualized by the model.

Regarding the data in Figure 11, we would like to point out that these show the relative changes, so even if it is true that the transition from WATCH to ALERT occurs mostly in autumn, it is also worth to point out that the area under drought is overall smaller in autumn compared with summer (e.g. see previous Figures 7-10). Hence, overall, the new index shows that after the peak the area under drought reduces in size and its mostly constituted by ALERT (as expected), whereas in the previous version of the index there where still sub-areas that were under WATCH even when the drought was almost over.

We revised the discussion section of the manuscript to improve the analysis on the depiction of a drought evolution according to the new version of the index.

Finally, the Iberian Peninsula was indeed affected by a METEOROLOGICAL drought roughly between October 2004 and August 2005, as the reviewer correctly points out. However, our index captures also the propagation of the drought into soil moisture and vegetation, and it is likely that the vegetation in August, after a full hydrological year under drought, did not recover immediately but remained under drought conditions after that date and into autumn (when significant rainfall arrived in the Mediterranean). This case study actually highlights quite well one misinterpretation of the old CDI version, which reports a recovery in August due to the return to normal conditions of SPI, even if fAPAR anomalies are still strongly negative. In this case, the increase in ALERT is compensated by the reduction in recovery classes, not reported in Figure 11 but visible in Figure 8 in the map for August.

We separated the discussion of the thee major droughts, in order to provide a better description of the evolution of the events. Additionally, in order to improve the analysis on the temporal evolution of the analysed events, we expanded section 2.4 with some quantitative information derived from EUROSTAT yield data. We used these information to further support the improved performance of the revised index.

4. Comparison with other hybrid indices

In the case of drought is difficult to know when an event starts or ends. The classification of drought is also a challenging task.
Therefore, a validation of CDI or another drought indicator is challenging. However, in my opinion it is not enough to evaluate an indicator without an exhaustive comparison with other indicators (multiscalar indicators, vegetation indicators, among others). A comparison of the new version with the previous version of the same index seems to be not sufficient, namely in the case of a product that is produced and disseminate operationally.

As the reviewer correctly points out, the absence of reference information for the start/end of a drought makes validating the performance of the index quite challenging. This is why we focused on highlighting how the new version of the index is an improvement of the previous one, rather than on an absolute validation of the index. Validation of the original version of the index has been done in previous studies by comparing agricultural yields of regions dominated by croplands with the CDI. Since the proposed changes do not alter completely the index behaviour, we can expect that the new method will give more or less similar results.

Similarly, we do not consider a comparison with other indicators as a valid approach to highlight how the new version improves over the previous one, since no other index can be reasonably assumed as a target reference.

In the new version of the manuscript we highlight this goal in the introduction, and we also introduced other alternative independent sources of information on the impacts of drought on vegetated land (e.g. EUROSTAT yield) in order to support the fact that more extended ALERT areas are expected during events with documented large impacts in yield.

[revised manuscript text omitted]